# The impact of brain iron accumulation on cognition: A systematic review

**Holly Spence** *, **Chris J. McNeil, Gordon D. Waiter**

Aberdeen Biomedical Imaging Centre, Institute of Medical Sciences, University of Aberdeen, Aberdeen, United Kingdom

* h.spence.19@abdn.ac.uk

## Abstract

Iron is involved in many processes in the brain including, myelin generation, mitochondrial function, synthesis of ATP and DNA and the cycling of neurotransmitters. Disruption of normal iron homeostasis can result in iron accumulation in the brain, which in turn can partake in interactions which amplify oxidative damage. The development of MRI techniques for quantifying brain iron has allowed for the characterisation of the impact that brain iron has on cognition and neurodegeneration. This review uses a systematic approach to collate and evaluate the current literature which explores the relationship between brain iron and cognition. The following databases were searched in keeping with a predetermined inclusion criterion: Embase Ovid, PubMed and PsychInfo (from inception to 31$^{st}$ March 2020). The included studies were assessed for study characteristics and quality and their results were extracted and summarised. This review identified 41 human studies of varying design, which statistically assessed the relationship between brain iron and cognition. The most consistently reported interactions were in the Caudate nuclei, where increasing iron correlated poorer memory and general cognitive performance in adulthood. There were also consistent reports of a correlation between increased Hippocampal and Thalamic iron and poorer memory performance, as well as, between iron in the Putamen and Globus Pallidus and general cognition. We conclude that there is consistent evidence that brain iron is detrimental to cognitive health, however, more longitudinal studies will be required to fully understand this relationship and to determine whether iron occurs as a primary cause or secondary effect of cognitive decline.

## Introduction

Iron has many biological roles including the cycling of neurotransmitters, enzyme and mitochondrial function, ATP and DNA synthesis and myelin generation [1–4]. In the healthy human adult brain, the total concentration of iron is around 0–200μg per gram of tissue, typically being lower in the White Matter (WM) and cortical Grey Matter (GM) (<60 μg per gram) [2]. 90% of brain iron is stored in ferritin with only 0.05% of brain iron being present in the labile iron pool [5]. In healthy aging, iron accumulates heterogeneously in specific regions of the brain, bound mainly to ferritin and neuromelanin [6] and largely located in the deep

**Funding:** Principal Grant Holder: GW Funder: The Roland Sutton Academic Trust https://www.abdn.ac.uk/ims/research/abic/roland-sutton-academic-trust-1427.php Sponsers only provided financial support.

**Competing interests:** The authors have declared that no competing interests exist.

GM nuclei [7–10]. There is a rapid increase in iron accumulation (different depending on brain region) from birth up until around 20 years old, at which point the accumulation rate slows in some regions, reaching a plateau in middle age and increasing again after 60 years old [1, 7]. Due to this relationship with age, brain iron has been the focus of many studies finding associations between regional brain iron levels and age-related cognitive decline, as well as several neurodegenerative diseases [4, 6].

Several theories as to the role of brain iron in cognitive decline have been suggested. Many of these mechanisms revolve around the ability of iron to induce oxidative stress via Fenton's reaction [11]. During Fenton's reaction, excess iron reacts with reactive oxygen species (ROS), such as hydrogen peroxide to produce highly reactive OH radicals which can in turn induce iron release from mitochondrial iron-sulphur cluster proteins and iron storage proteins. Released iron can then undergo Fenton's reaction, amplifying ROS generation [6, 12]. When ROS and free radicals generated via Fenton's reaction exceed the antioxidant capacity of brain cells, oxidative stress is induced leading to loss of DNA integrity, lipid peroxidation, mitochondrial dysfunction, protein misfolding and ultimately neuronal cell death. This oxidative stress is thought to be exacerbated by the induction of neuroinflammation. Upregulation of HO-1 in glia is also thought to contribute to neurodegeneration as prolonged action may be involved in iron sequestration, intracellular stress and mitochondrial insufficiency [6, 13–15]. Another potential mechanism by which brain iron levels could influence cognitive decline/neurodegeneration is Ferroptosis. This is an iron-dependent necrosis mechanism which is characterised by shrunken mitochondria with increased density and outer membrane rupture [16].

In terms of the mechanism by which iron accumulates, it has been shown in several studies that participants with Parkinson's disease (PD) have an increased permeability of the blood brain barrier (BBB) and upregulation of iron transporters such as Lactotransferrin [5, 12, 17]. This would allow for the increased uptake of iron into the brain and may account for the increase of iron accumulation in the brain in neurodegenerative disease. Furthermore, in diseases such as Alzheimer's disease (AD), PD and prion disease, iron is shown to associate with protein aggregates and in the case of Amyloid beta, it is thought that iron plays a role in the toxicity of these protein aggregates [18–20]. Although these theories have been proposed, the full extent of the role of iron in cognitive decline and neurodegeneration remains unclear.

Although iron status measurement has been possible for many years, the emergence of novel techniques in magnetic resonance imaging have allowed for the specific, non-invasive measurement of brain iron. Techniques such as Susceptibility Weighted Imaging (SWI), R2* relaxation time and Quantitative Susceptibility Mapping (QSM) make use of the magnetic properties of iron in order to map the spatial distribution of iron in the brain from magnitude and phase images [21]. QSM is considered the most sensitive and specific technique for measuring iron in the brain non-invasively [22] and the ability of QSM to accurately measure brain iron has been validated in several post-mortem studies [23, 24]. It is hoped that such measures of brain iron will allow for further elucidation of the brain iron accumulation patterns and their relationship with cognitive decline and neurodegeneration.

This review will discuss the relationships between brain iron and cognition elucidated in human studies across a wide age range; in healthy adults as well as, in individuals with diseases including PD, AD, Type 2 Diabetes Mellitus with cognitive impairment, mild cognitive impairment (MCI) and Multiple Sclerosis (MS). We aim to present the current understanding of regional brain iron accumulation patterns and their relation to cognitive performance outcomes, in order to gain a greater understanding of the potential mechanisms underlying this iron-cognition relationship. We hypothesise that age-related regional increase in brain iron levels will correlate with impairment of specific regional cognitive function.

## Methods

The PRISMA statement recommendations for systematic review were followed in this systematic review in order to provide high quality reporting [25, 26].

### Information sources and eligibility criteria

A systematic electronic search strategy was generated at the start of this study. Electronic searching was carried out on 31st March 2020 using the following electronic databases: Embase Ovid (1974–31 March 2020), PubMed (Inception– 31 March 2020) and Psych Info (1806–31 March 2020). Studies were assessed for adherence to pre-determined inclusion/exclusion criteria detailed below.

### Inclusion criteria

Studies were included in this review if they reported on the following:

1. Human Studies measuring Brain iron level AND cognition

2. Statistical comparison of brain iron and cognitive performance

3. Published in English Language AND the full text was available

### Exclusion criteria

Studies were excluded from this review if they were:

1. Animal studies

2. Measuring only systemic iron status (no measure of brain specific iron levels)

3. Single case studies, Reviews, protocols editorials or conference abstract

4. Studies investigating effects of maternal iron on offspring cognition

### Search strategy

The full search strategies used for this review are detailed in Table 1.

Table 1. Search strategy.

| Database | Search terms |
|---|---|
| Embase Ovid (1974–31 March 2020) | (("cognitive".ti OR "neurocognitive".ti OR "cognitive decline".ti OR "mental deterioration".ti OR "cognition".ti OR "brain function".ti OR "brain health".ti OR "cognitive ability".ti OR "cognitive health".ti OR "cognitive function".ti OR "neurological health".ti OR "neurological".ti) AND ("iron".ti OR "Fe".ti OR "ferric".ti OR "ferrous".ti OR "ferritin".ti OR "transferrin".ti OR "TfR".ti)) |
| Pubmed (Inception—31 March 2020) | (("cognitive"[title] OR "neurocognitive"[title] OR "cognitive decline"[title] OR "mental deterioration"[title] OR "cognition"[title] OR "brain function"[title] OR "brain health"[title] OR "cognitive ability"[title] OR "cognitive health"[title] OR "cognitive function"[title] OR "neurological health"[title] OR "neurological"[title] AND ("iron"[title] OR "Fe"[title] OR "ferric"[title] OR "ferrous"[title] OR "ferritin"[title] OR "transferrin"[title] OR "TfR"[title])) |
| Psycinfo (1806–31 March 2020) | (("cognitive".ti OR "neurocognitive".ti OR "cognitive decline".ti OR "mental deterioration".ti OR "cognition".ti OR "brain function".ti OR "brain health".ti OR "cognitive ability".ti OR "cognitive health".ti OR "cognitive function".ti OR "neurological health".ti OR "neurological".ti) AND ("iron".ti OR "Fe".ti OR "ferric".ti OR "ferrous".ti OR "ferritin".ti OR "transferrin".ti OR "TfR".ti)) |

## Study selection

All studies found in the electronic search were assessed for their eligibility for inclusion in this review by Holly Spence. Studies were included if they met all the inclusion criteria and included studies had their referenced papers reviewed for eligibility for inclusion. The included studies ultimately consisted of published articles and theses only.

## Synthesis of results

The following study characteristics were extracted from each study for assessment of study quality and study comparison: Number of participants; participant gender ratio; participant average age; type of study design; measures of cognition used; measures of brain iron used; statistical methods used. Results which were statistically significant ($p < 0.05$) were extracted and summarised from each study.

## Quality assessment

Each study which satisfied the inclusion and exclusion criteria was assessed for quality via a 10-point based system using the following 10 criteria: (1) Does the study have a clearly defined research objective? (2) Does the study adequately describe the inclusion/exclusion criteria? (3) Is the sample size adequate? (4) Does the study report on the population parameters/demographics? (5) Does the study report detail on appropriate assessment of Cognition? (6) Does the study report detail of the assessment of iron? (7) Does the study provide an appropriate control group? (8) Does the study apply the appropriate statistical analyses? (9) Does the study adequately report the strength of results? (10) Do the authors report on the limitations of their study?

# Results

## Study selection

The electronic search of Embase Ovid, PubMed and PsychInfo yielded 643 citations in total. After duplicates were removed, 411 studies remained for screening. Once these studies were screened, 141 non-human studies were excluded, 4 papers were excluded due to being unavailable in English and 106 reviews, editorials and conference abstracts were removed. In total 6 studies were excluded due to being single case studies, 60 were excluded for not measuring brain iron and 6 were excluded for not measuring cognition. Finally, 24 studies did not assess statistically the relationship between iron and cognition and so were excluded, as well as 3 studies which measured only the effects of maternal iron on offspring cognition. A total of 28 studies remained for reference screening. After references were reviewed, a further 13 eligible studies were obtained. A total of 41 studies were therefore included in this review. The full details of the study selection process are outlined in Fig 1.

## Study characteristics

Study characteristics were collated and are presented in Tables 2 and 3; with Table 2 presenting details on overall study design and Table 3 presenting details on participants and study groups

## Quality assessment

All included studies were assessed for quality using a 10-point-based scoring system and each score was converted to a % Quality Score (QS). The quality scores for all studies can be seen in

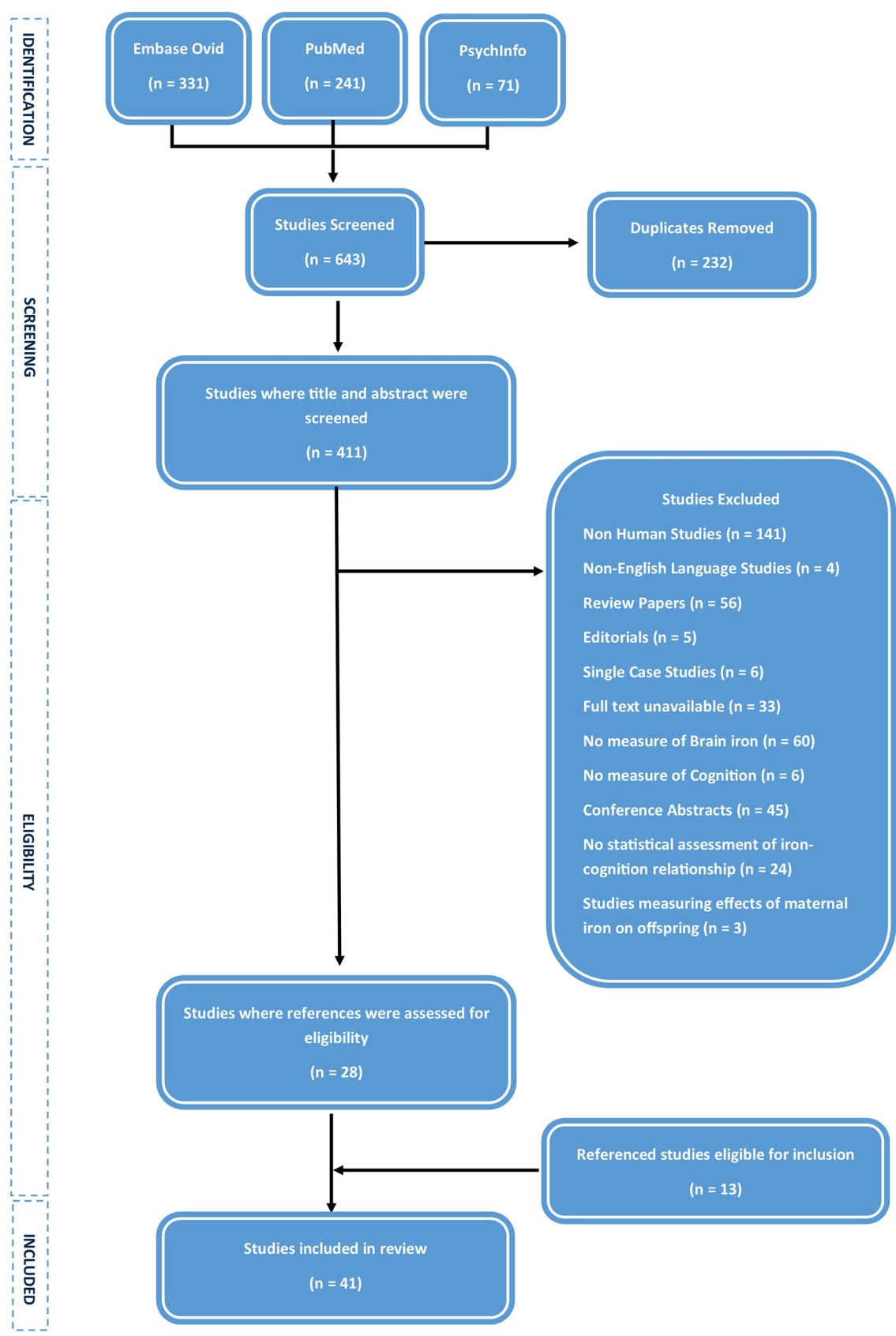

**Fig 1. Flowchart detailing the study selection procedure.**

**Table 2. Study characteristics.**

| Study Reference | Total Participants (n) | Study Design | Method of Cognitive Assessment | Method of Iron Assessment | Statistical Methods Used |
|---|---|---|---|---|---|
| **Ayton et al., 2019** [27] | 209 | Cross-sectional study | Word List Memory; Word List Recall and Word List Recognition from the procedures established by the CERAD; immediate and delayed recall of Story A from the Logical Memory subtest of the Wechsler Memory Scale- Revised; immediate and delayed recall of the East Boston Story; 15-item Boston Naming Test; Verbal Fluency; 15-item word reading test; Digit Span Forward; DSB; Digit Ordering; SDMT; Stroop colour naming; Stroop word reading; number comparisons; Judgement of Line Orientation; 16-item version of the Standard Progressive Matrices; Composite scores computed for each cognitive domain and for global cognition using standardized composite z-scores | Post-mortem instrumental neutron activation analysis | Multiple regression models; Bonferroni adjustment for multiple comparisons; Mixed-effects linear models of cognitive composite scores; Mediation analysis |
| **Bartzokis et al., 2011** [28] | 63 | Cross-sectional study | CVLT for verbal learning and memory; ACT for working memory; Digit Span subtest of WAIS III for attention span measure; TMT-A and B for processing speed, Digit Symbol subtest from WAIS-R to measure psychomotor speed | MRI (1.5/0.5T) with FDRI | Multiple regression analysis; Pearson's correlations; Post hoc Fisher's z-transformed values; Principle components analysis; Bonferroni correction for multiple comparisons |
| **Blasco et al., 2014** [29] | 43 | Cross-sectional study | WAIS-III; TMT; ROCF test; Stroop neuropsychological screening test; Iowa gambling task | MRI (1.5T) with R2* Relaxometry | Chi-squared for categorical variables; Students t test for quantitative variables; Mann-Whitney U test for non-normal distributed variables; Nonparametric spearman analysis; Multivariate linear regression modelling controlling for age, sex and BMI, Receiver operating characteristic analyses |
| **Blasco et al., 2017** [30] | 35 | Longitudinal study | ROCFT; TMT A and B; Verbal Fluency | MRI (1.5T) R2* Relaxometry | Voxel wise non-parametric permutation inference with 5000 randomised iterations to evaluate association of age and obesity variations in R2*; Continuous variables analysed with median and quartiles; Categorical variables analysed with frequencies; Non-parametric analyses considering non-normal distribution due to small sample size; Mann-Whitney U test for differences in study groups; Wilcoxon test for longitudinal intrasubject differences; Correlations assessed by Spearman rho analysis; Clustering analysis; Multivariate linear regression |
| **Carpenter et al., 2016** [31] | 39 | Longitudinal Study | DAS to measure overall general intelligence and cluster scores for verbal ability; non-verbal reasoning and spatial ability and reported as age-based standard scores | MRI (3T) with QSM | Mixed multiple regression models with age at scan as a covariate |

*(Continued)*

**Table 2.** (*Continued*)

| Study Reference | Total Participants (n) | Study Design | Method of Cognitive Assessment | Method of Iron Assessment | Statistical Methods Used |
|---|---|---|---|---|---|
| **Chen et al., 2018** [32] | 27 | Longitudinal Study | NIH toolbox for cognition batteries; Executive function and attention; episodic memory; working memory and language processing summary scores | MRI (3T) with SWI and QSM | Unconditional logistic regression to compare ethnicity and education; Fishers test to assess ethnicity distribution between groups; Linear mixed modelling for longitudinal analyses; Compound symmetry covariance structure for within-subject correlations; Pearson correlation coefficients; Bonferroni method for multiple testing correction |
| **Darki et al., 2016** [33] | 46 | Cross-sectional study | Visuo-spatial working memory task | MRI (3T) with QSM | Students t test; ANOVAs; Bonferroni Correction for multiple comparisons |
| **Daugherty, 2014** [34] | 89 | Longitudinal Study | Listening span; n-back for digits; SPART; n-back for objects; Virtual Morris water maze | MRI (4T) with SWI | Latent growth curve model analyses; Latent change score model analyses; Missing data was estimated with full information maximum likelihood in Mplus; ANCOVA for adjusting volume to cranial size; Bootstrapping with bias correction for sample size, simple effects models; Bonferroni correction for multiple comparisons; Chi-squared statistic; Root mean square error of approximation (RMSEA); Comparative fit; Tucker-Lewis fit; Standardised root mean residual (SRMR) |
| **Daugherty, Haacke and Raz, 2015** [35] | 125 | Longitudinal study | Working memory (Listening span; Verbal N-back task; SPART; Non-verbal N-back task); Episodic memory (Logical memory subset of Wechsler Memory scale-revised) | MRI (4T) with SWI and $R2^*$ relaxometry | Longitudinal structural equation modelling; Univariate distributions; Tukey's boxplots; Z-scores; Latent change score models; parallel process models; Bonferroni correction for multiple comparisons; Models bootstrapped with bias correction; Normal theory weighted chai squared test; root mean square error of approximation; Comparative fit index; Standardised root mean residual; Weighted root mean square residual, Reverse effects models |
| **Daugherty et al., 2019** [36] | 183 | Cross-sectional study | CES-D; MMSE; Verbal Fluency and Colour Word Interference Subtests of the D-KEFS | MRI (3T) SWI | z test; chi squared tests; comparative fit index (excellent fit when 0.90); root mean square of error approximation (good fit when <0.08); standardised root mean residual (good fit when <0.08) |
| **Ding et al., 2009** [37] | 50 | Cross-sectional study | MMSE | MRI (1.5T) with phase imaging | One-way ANOVA with LSD post hoc test; Students t-test; Mann-Whitney U test; Paired-sample t-test; Partial Spearman rank correlation coefficient |
| **Du et al., 2018** [38] | 60 | Case-Control Study | MMSE and MoCA | MRI (3T) QSM | Intraclass correlation coefficient for interobserver error (<0.4 Poor, 0.4–0.59 Fair, 0.6–0.74 Good, >0.74 Excellent); Pearson partial correlation for correlation analyses; Paired t test and Student t test |

(*Continued*)

**Table 2.** (Continued)

| Study Reference | Total Participants (n) | Study Design | Method of Cognitive Assessment | Method of Iron Assessment | Statistical Methods Used |
|---|---|---|---|---|---|
| **Fujiwara et al., 2017** [39] | 67 | Case-Control Study | Verbal and visual memory tests; SRT; SPART; Info processing speed/ working memory tests (SDMT and PASAT); Phonemic fluency test (WLG) | MRI (4.7T) R2* Relaxometry and QSM | Cognitive scores and imaged parameters compared using ANCOVAs; Hierarchic Linear regression models |
| **Gao et al., 2017** [40] | 90 | Case-Control Study | MMSE and MoCA | MRI (3T) SWI | Chi-squared tests for categorical variables; One-way ANOVA for comparison of groups using Fisher's LSD posthoc test; Pearson correlation coefficient used to analyse relationship between quantitative variables; Serum ferritin had large variance so log10 serum ferritin was used in correlation analysis |
| **Ge et al., 2007** [41] | 31 | Cross-sectional study | WAIS-III; Digit Symbol-coding; ROCFT; D-KEFS; verbal fluency test; CVLT; WAIS-III; SDMT; D-KEFS Colour Word Interference Test; PASAT | MRI (3T) with magnetic field correlation | Least Squares Regression; Pearson Correlations Coefficients |
| **Ghadery et al., 2015** [42] | 336 | Cross-sectional study | Immediate memory recall and learning ability (Lern und Gedächtnis Test, word and digit association tasks and story recall, trail and design recall); Executive function (WCST, TMT-B and DSB, (part of the WAIS-III)); Psychomotor speed (Purdue Pegboard test). Each of 3 cog function measures was summarised by z scores and global cog function calculated as mean of 3 cognitive function measures | MRI (3T) R2* Relaxometry | Kolmogorov-Smirnov test; ANOVA with Kruskal-Wallis test for normally dist.; Difference in proportions chi-squared; Regression analyses; Family structure added to models as random effect; Mediation models with bootstrapping; All models adjusted for potential cofounders of age, sex, education, hypertension, diabetes, cardiac disease; Variation inflation factor for multicollinearity; Benjamin Hochberg false discovery rate method for multiple testing correction |
| **Haller et al., 2010** [43] | 102 | Cross-sectional study | MMSE; verbal fluency; digit span forward; shapes tests | MRI (3T) SWI | ANOVA with post hoc pairwise Tukey multiple comparison test for parametric demographic measures; Non parametric data analysed by Kruskal-Wallis group tests with post hoc pairwise Dunn multiple comparison test; Cerebral microhaemorrhages differences were analysed by Mann-Whitney tests; Differences in iron deposition were assessed by ANOVAs with post hoc Bonferroni multiple comparison test for comparing controls and MCI groups |
| **Hect et al., 2018** [44] | 57 | Cross-sectional study | IQ test calculated from Kaufman Brief Intelligence Test (2nd edition); W-J III NU (visual matching and cross out subtests); WAIS III (Digit symbol test) | MRI (3T) R2* Relaxometry | General linear models for age differences in iron in relation to cognition; hierarchical linear regression models to identify unique and shared variance bootstrapped with bias-correction and Bonferroni correction for multiple comparisons |

(*Continued*)

**Table 2.** (Continued)

| Study Reference | Total Participants (n) | Study Design | Method of Cognitive Assessment | Method of Iron Assessment | Statistical Methods Used |
|---|---|---|---|---|---|
| **House et al., 2006** [45] | 23 | Cross-sectional study | CVLT to assess verbal learning and memory; MMSE; NART; DASS | MRI (1.5T) with R2 Relaxometry | Pearson product moment correlation coefficient; ANOVA; Two-tailed t tests; Chi-squared test; One-tailed t-test; ANCOVA with age as covariate; Partial Spearman rank correlation coefficient |
| **Kalpouzos et al., 2017** [46] | 37 | Cross-sectional study | Mental Imagery Memory task (Imagery then recall of scenes involving motion and involving no motion) | MRI (3T) with R2* Relaxometry | ANOVA; Partial correlations; ANCOVAs; Paired and 2-sample T-tests; Linear models; Cluster analyses; Bootstrapping analysis for small sample size |
| **Larsen et al., 2020** [1] | 818 | Longitudinal Study | Penn computerised neurocognitive battery (CNB) with 14 subtests assessing executive control; complex cognition; episodic memory; social cognition and motor speed | MRI with R2* Relaxometry | Interacquisition variability corrected using ComBat batch effect correction tool with age, sex, visit number and cognitive performance as covariates; Bonferroni correction for multiple comparisons; Generalised Additive Mixed Model for linear/non-linear age effects and cognitive effects; Bivariate smooth model and varying coefficient models; Bayesian information criterion for model selection; P values confirmed using parametric bootstrap likelihood ratio test |
| **Li et al., 2015** [47] | 132 | Cross-sectional study | Purdue-Pegboard-Test for manual dexterity and perceptual speed; Digit span test for verbal working memory; WCST to measure ability to display flexibility in face of changing rules; TMT-B for measuring executive functioning, psychomotor speed and visual scanning; Semantic Fluency Test; MMSE | MRI (3T) with QSM | Multiple Regression Modelling; Standardised z scoring susceptibility, demographic and behavioural variables; Factor analysis |
| **Lu et al., 2015** [48] | 76 | Cross-sectional study | MMSE | MRI (3T) with SWI | ANCOVA; correction for multiple comparisons by Levene's Test for Equality of Variances, Spearman Correlations for MMSE-Angle Radian value relationship |
| **Modica et al., 2015** [49] | 112 | Cross-sectional study | SDMT for visual information processing speed; 3 second interval PASAT for auditory information processing; correct sorts component of D-KEFS; total learning portion of the second edition CVLT and the total learning portion of the BVMT-R | MRI (3T) SWI | MS and controls cognition compared by One-way ANOVA; Z scores calculated for each cog test based on controls; Partial correlations controlling for age and education; Pearson correlations assessed between structure mean phase and volume of structure; Hierarchical linear regression analysed mean phase-cognitive test relationship |
| **Murakami et al., 2018** [50] | 49 | Cross-sectional study | MMSE; FAB for frontal lobe function; MRS for neurological disturbance | Brain-type Transferrin assessed via SDS-PAGE and PVL lectin staining | Parametric/non-parametric was assessed by Kolmogorov-Smirnov or Shapiro-Wilk method; Parametric variables assessed with Mean and SD; Students t test; Welch test; Multiple comparisons Dunnett's test; Pearson correlation coefficients |

*(Continued)*

**Table 2.** (*Continued*)

| Study Reference | Total Participants (n) | Study Design | Method of Cognitive Assessment | Method of Iron Assessment | Statistical Methods Used |
|---|---|---|---|---|---|
| **Penke et al., 2012** [51] | 143 | Longitudinal study | At 11 years old—MHT number 12 for general IQ; At 70 years old—MHT; At 72 years old—WAIS-III including symbol search, digit symbol, matrix reasoning, letter-number sequencing, DSB and block design; NART and WTAR | MRI (1.5T) with MCMxxxVI method (multispectral colouring modulation and variance identification) for iron quantification | Age was controlled for in all analyses; Total iron volume was standardised to brain volume for each subject to derive % of iron deposit in brain tissue; Tobit regressions with iron deposition as dependant variable to calculate censored correlations with cognition |
| **Pinter et al., 2015** [52] | 69 | Cross-sectional study | EDSS; Brief Battery of Neuropsychological Tests; SRT; 10/36-SPART; SDMT; PASAT; WLG; Composite z score to measure overall cognitive function | MRI (3T) with R2* relaxometry | Pearson Correlation; Point-biserial Correlation; Durbin-Watson-test; Variance Inflation Factor; Hierarchical regression models; Multivariate model including strongest predictors for overall cognitive function and subdomains to assess additive value of multiple MRI-parameters in predicting cognition |
| **Qin et al., 2011** [53] | 30 | Cross-sectional study | MMSE | MRI (3T) with R2* Relaxometry | Pearson correlation assay for Linear regression; two tailed t-test; Students-Newman-Keuls test for ANOVA; Linear correlation test |
| **Rodrigue et al., 2013** [54] | 113 | Cross-sectional study | Immediate and delayed recall measures from memory for names (W-J III NU) and logical memory tests (Wechsler Memory Scale Revised) | MRI (1.5T) with T2* relaxometry | Structural equation modelling with latent variables; All memory and anatomical measures log transformed to alleviate skew; Bootstrapping to combat modest sample size with bias correction (500 iterations of whole sample); Chi-squared statistic; Root mean square error of approximation (RMSEA); Comparative fit; Tucker-Lewis fit indices; Standard root mean residual (SRMR); Akaike and sample-size adjusted Bayesian information criteria; James and Brett method for evaluation of indirect effects |
| **Rodrigue et al., 2020** [55] | 166 | Cross-sectional study | Executive function (Subsets of the D-KEFS—Verbal fluency, TMT and Colour word interference test); and Subsets of the WCST and Working Memory (WAIS-IV subtests, Digit span forward backward, Listening Span Task); Both functions are standardised together to form z scores | MRI (3T) with R2* Relaxometry | General linear models, Age, iron, age x iron interaction, whole brain CBF, sex and task response time were used as between subjects 2nd level covariates for linear modelling; All covariate mean centred to avoid bias in regression coefficients from multicollinearity; Cluster corrections calculated using non-parametric mapping toolbox |
| **Salami et al., 2018** [56] | 42 | Cross-sectional study | Purdue Pegboard Task | MRI (3T) with R2* Relaxometry and QSM | Two sample t-test; Multivariate model selection with sex as covariate; Partial correlations between iron content and connectivity; MANCOVA; Comparison between correlations conducted using Steigers z-test |

(*Continued*)

**Table 2.** (*Continued*)

| Study Reference | Total Participants (n) | Study Design | Method of Cognitive Assessment | Method of Iron Assessment | Statistical Methods Used |
|---|---|---|---|---|---|
| **Schmalbrock et al., 2016** [57] | 29 | Cross-sectional study | MMSE; WTAR and the brief repeatable battery to assess general cognition; computerised versions of Flanker and Stroop tasks to assess inhibitory control | MRI (7T) with QSM and R2* relaxometry | z score deviation of >2.5SDs from mean were classed as outliers; normality tested via Shapiro-Wilk test; corrected skewed data with square root transformation; Pearson correlations; subtracted linear regression fit of QSM with age and EDSS with disease duration from measured data to control for these influences; semi partial correlations |
| **Smith et al., 2010** [58] | 20 | Cross-sectional study | CDR | Histochemistry (7% Potassium ferrocyanide in 3% HCL visualised by treating with 0.75mg/ml 3-3-diaminobenzidine in tris buffer with H2O) | Students t test |
| **Steiger et al., 2016** [59] | 62 | Cross-sectional study | Verbal Learning and Memory test | MRI (3T) with R2* relaxometry | Two sample t-test; Familywise error correction for multiple comparisons; Whole-brain linear regression modelling (Scores were used as individual regressors on the probability maps) |
| **Sullivan et al., 2009** [60] | 10 | Retrospective Study | MMSE; Mattis Dementia Scale (5 subtests–memory, arithmetic, construction, conceptualisation and initiation/preservation); Digit symbol test; Fine Finger Movement Test to assess upper limb speed; Two choice Task to assess reaction time and movement time | MRI (1.5T and 3T) with Field Dependent R2 increase | Pearson product-moment correlations for relationships between iron and cognitive tests; due to small sample size, parametric correlations were confirmed with non-parametric spearman rank order tests |
| **Sun et al., 2017** [61] | 39 | Cross-sectional study | Attention-executive function (Chinese modified version of TMT; modified version of Stroop Colour-Word Test and Verbal Fluency Test), Memory Function (Chinese modified version of AVLT for short and long delay free recall and ROCF delayed recall test); Language function (Boson Naming Test) and Visuospatial Function (ROCF copy test); Z score calculated for each function and a composite z score for all functions | MRI (3T) with QSM | Independent 2 sample t-test; Chi squared for calculating gender heterogeneity between groups; Non-normally distributed data was compared using Mann-Whitney U test; Inter-rater reliability among all regions = 0.947 for ROI segmentation; Correlation analyses with age and gender as covariates for z score-iron correlations |
| **Thomas et al., 2020** [62] | 137 | Cross-sectional study | MoCA; MDS- UPDRS- III; REM-sleep behaviour disorder score; Sense of smell; Depression; visual acuity as assessed by LogMAR; Colour vision D15 test; Contrast sensitivity using Pelli-Robson chart; Cats and Dogs task 25 and Biological Motion test | MRI (3T) with QSM | Age and total intracranial volume controlled for in imaging analyses as nuisance covariates; Permutation based regression analyses; Wilcoxon rank-sum tests; QSM values age-corrected using covariance method |

(*Continued*)

**Table 2.** (Continued)

| Study Reference | Total Participants (n) | Study Design | Method of Cognitive Assessment | Method of Iron Assessment | Statistical Methods Used |
|---|---|---|---|---|---|
| **Valdes-Hernandez et al., 2015** [63] | 676 | Cross-sectional study | Fluid intelligence (g-fluid) consisting of; Digit symbol substitution test, DSB, symbol search, letter-number sequencing, block design and matrix reasoning. General processing speed (g-speed) consisting of; simple reaction time and choice reaction time, inspection time test, digit symbol substitution and symbol search. General memory (g-memory) consisting of logical memory total, verbal paired associates (both at total, immediate and delayed recall) and spatial span total score, letter-number sequencing and DSB. | MRI (1.5T) T1/T2*W | Total and regional iron and WMH volumes were standardised and presented as % of intracranial volume, age was added as covariate of all models; volumes of iron and WMH were positively skewed and so were log transformed prior to analysis; Multivariate and Bivariate regression models were performed |
| **Van Bergen et al., 2016** [64] | 37 | Cross-sectional study | MMSE, MoCA, verbal learning and memory test; Wechsler Memory Scale; Boston naming test; TMT-A and B | MRI (7T) QSM | One-Way MANCOVA for differences between groups; Cohens d for effect size; Spearman's rho |
| **Wang et al., 2013** [65] | 60 | Cross-sectional study | MMSE; CDR | MRI (3T) with SWI | Pearson correlation coefficients; ANOVA; Fish-Least significant difference (LSD) test |
| **Yang et al., 2018** [66] | 90 | Case-Control Study | AVLT; complex figure test; digit symbol coding test; digit span test; verbal fluency test; TMT-A and B | MRI (3T) SWI | Distribution assessed using Kolmogorov-Smirnov test; ANOVAs for normally distributed continuous data and LSD test used for post hoc analysis; Kruskal-Wallis H test used for non-normally distributed or unequal variances data and Mann-Whitney U test was used for posthoc analysis with sig level adjusted by Bonferroni correction; Chi squared test to compare proportions; Independent 2 sample t test used to assess diabetes duration |

CVLT = California Verbal Learning Test; ACT = Auditory Consonant Trigrams; TMT-A and -B = Trail Making Task -A and -B; WAIS-III = Wechsler Adult Intelligence Scaled Third Edition; ROCF = Rey-Osterrieth Complex Figure; DAS = Differential Abilities Scale; CES-D = Centre for Epidemiologic Studies Depression Scale; MMSE = Mini Mental State Examination; D-KEFS = Delis-Kaplan Executive Function Test Battery; MoCA = Montreal cognitive assessment; SRT = Selective Reminding Task; SPART = Spatial Recall Test; SDMT = Symbol Digit Modalities Test; PASAT = Paced Auditory Serial Addition Test; DSB = Digit Span Backwards; W-J III NU = Woodcock-Johnson III Normative Update for processing speed; NART = National Adult Reading Test; DASS = Self-evaluated stress, anxiety and depression questionnaire; WCST = Wisconsin Card Sorting Test; BVMT-R = Brief Visuospatial Memory Test–Revised; MRS = Modified Rankin Scale; MHT = Moray House Test; WTAR = Wechsler Test of Adult Reading; EDSS = Expanded Disability Status Score; WLG = Word List Generation; CDR = Clinical Dementia Rating Scale; AVLT = Auditory Verbal Learning Test; MDS-UPDRS = 2 Year Risk of cognitive decline score made up of Movement Disorder Society Unified Parkinson's Disease Rating Scale motor part 3.

Table 4. 26 of the 41 assessed studies were of high quality (QS>90%), 14 were of very good quality (QS of 80%-90%) and 1 study was of good quality (QS of 70%-80%).

## Summary of results

The key findings relating to the brain iron-cognition relationship were extracted from each study in this review and are summarised in Table 5. 11 of the reviewed papers showed a significant relationship between whole brain iron concentration and measures of cognition

**Table 3. Participant characteristics.**

| Study Reference | Total Participants (n) | Participants (Control group) | | | | Participants (Study group 1) | | | | Participants (Study group 2) | | | | Participants (Study group 3) | | | |
|---|---|---|---|---|---|---|---|---|---|---|---|---|---|---|---|---|---|
| | | n | Gender Ratio (Men:Women) | Mean Age at baseline (years) | Group | n | Gender Ratio (Men:Women) | Mean Age at baseline (years) | Group | n | Gender Ratio (Men:Women) | Mean Age at baseline (years) | Group | n | Gender Ratio (Men:Women) | Mean Age at baseline (Mean years ± SD) | Group |
| **Ayton et al., 2019** [27] | 209 | 69 | 46:23 | 87.8 ± 6.0 | Cognitively normal subjects with low AD pathology post-mortem | 14 | 07:07 | 92.9 ± 4.5 | AD patients with low pathology post-mortem | 71 | 51:20 | 89.9 ± 6.5 | Cognitively normal subjects with high AD pathology post-mortem | 55 | 36:19 | 90.7 ± 5.1 | AD patients with high AD pathology post-mortem |
| **Bartzokis et al., 2011** [28] | 63 | - | - | - | - | 63 | 33:30 | 67.0 ± 6.1 | Healthy, Cognitively normal Adults | - | - | - | - | - | - | - | - |
| **Blasco et al., 2014** [29] | 43 | 20 | 10:10 | 48.8 ± 9.5 | Healthy Adult age and sex matched non-obese controls | 23 | 10:13 | 50.4 ± 7.7 | Middle aged Obese Subjects with type 2 Diabetes | - | - | - | - | - | - | - | - |
| **Blasco et al., 2017** [30] | 35 | 18 | 10:08 | (39–56.25) No mean given | Age and sex matched healthy non-obese adult controls | 17 | 06:11 | (48–58) No mean given | Obese adults | - | - | - | - | - | - | - | - |
| **Carpenter et al., 2016** [31] | 39 | - | - | - | - | 39 | 17:22 | 9.51 ± 1.25 | Healthy Children | - | - | - | - | - | - | - | - |
| **Chen et al., 2018** [32] | 27 | 13 | 00:13 | 68.2 ± 6.1 | Age matched women without breast cancer | 14 | 00:14 | 66.3 ± 5.3 | Women aged 60+ with breast cancer having adjuvant chemotherapy | - | - | - | - | - | - | - | - |
| **Darki et al., 2016** [33] | 46 | - | - | - | - | 25 | 16:09 | 29.1 ± 4.5 | Healthy Adults | 21 | 12:08 | 6.73 ± 0.27 | Healthy Children | - | - | - | - |
| **Daugherty, 2014** [34] | 89 | - | - | - | - | 89 | 26:63 | 55.18 ± 12.85 | Healthy Adults | - | - | - | - | - | - | - | - |
| **Daugherty, Haacke and Raz, 2015** [35] | 125 | - | - | - | - | 125 | 37:88 | 52.53 ± 14.91 | Healthy adults assessed at baseline | 78 | 24:54 | 56.87 ± 12.82 | Heathy adults followed up after 2 years | - | - | - | - |
| **Daugherty et al., 2019** [36] | 183 | - | - | - | - | 183 | 76:107 | 53.68 ± 18.96 | Health Adults | - | - | - | - | - | - | - | - |
| **Ding et al., 2009** [37] | 50 | 24 | 09:15 | 69.40 ± 11.38 | Healthy Age-matched controls | 26 | 08:18 | 70.96 ± 8.55 | AD patients | - | - | - | - | - | - | - | - |
| **Du et al., 2018** [38] | 60 | 30 | 10:20 | 66.2 ± 7.8 | Healthy Adult Controls | 30 | 09:21 | 68.3 ± 6.6 | Adults with mild to moderate Alzheimer's Disease | - | - | - | - | - | - | - | - |
| **Fujiwara et al., 2017** [39] | 67 | 27 | 09:18 | 47.51 ± 10.09 | Healthy Adult Controls | 40 | 13:27 | 49.08 ± 10.03 | Adult Multiple Sclerosis Patients | - | - | - | - | - | - | - | - |

(Continued)

Table 3. (Continued)

| Study Reference | Total Participants (n) | Participants (Control group) | | | | Participants (Study group 1) | | | | Participants (Study group 2) | | | | Participants (Study group 3) | | | |
|---|---|---|---|---|---|---|---|---|---|---|---|---|---|---|---|---|---|
| | | n | Gender Ratio (Men: Women) | Mean Age at baseline (years) | Group | n | Gender Ratio (Men: Women) | Mean Age at baseline (years) | Group | n | Gender Ratio (Men: Women) | Mean Age at baseline (years) | Group | n | Gender Ratio (Men: Women) | Mean Age at baseline (Mean years ± SD) | Group |
| Gao et al., 2017 [40] | 90 | 30 | 17:13 | 72.86 ± 5.75 | Age, Sex and Education matched healthy adult controls | 30 | 13:17 | 75.2 ± 5.75 | Adults with Mild Cognitive Impairment | 30 | 18:12 | 74.83 ± 4.52 | Adults with Alzheimer's Disease | - | - | - | - |
| Ge et al., 2007 [41] | 31 | 14 | 05:09 | 39 | Healthy adult controls | 17 | 03:14 | 44 | Relapsing-Remitting Multiple Sclerosis Patients | - | - | - | - | - | - | - | - |
| Ghadery et al., 2015 [42] | 336 | 336 | 132:204 | median age 67 (55–72) | Healthy Adult Participants | - | - | - | - | - | - | - | - | - | - | - | - |
| Haller et al., 2010 [43] | 102 | 35 | 09:26 | 63.7 ± 5.1 | Healthy adult controls | 40 | 13:27 | 65.4 ± 5.4 | Adults with stable mild cognitive impairment | 27 | 15:12 | 64.4 ± 4.6 | Adults with progressive mild cognitive impairment | - | - | - | - |
| Hect et al., 2018 [44] | 57 | - | - | - | - | 57 | 19:38 | 12.5 ± 2.36 | Healthy Children and Adolescents | - | - | - | - | - | - | - | - |
| House et al., 2006 [45] | 23 | 11 | 03:08 | 70.7 ± 6.9 | Healthy Adult Controls | 6 | 02:04 | 69.2 ± 6.8 | Elderly participants with memory complaints but no objective cognitive impairment | 6 | 01:05 | 75.5 ± 8.8 | Elderly participants with memory complaints and objective cognitive impairment | - | - | - | - |
| Kalpouzos et al., 2017 [46] | 37 | - | - | - | - | 22 | 10:12 | 36.8 ± 4.3 | Healthy Younger Adults | 15 | 08:07 | 69.7 ± 2.7 | Healthy Older Adults | - | - | - | - |
| Larsen et al., 2020 [1] | 818 | - | - | - | - | 818 | 389:429 | 14.84 ± 3.57 | Healthy Adolescents | - | - | - | - | - | - | - | - |
| Li et al., 2015 [47] | 132 | - | - | - | - | 132 | 54:78 | 64.5 ± 10.64 | Healthy, Cognitively normal Adults | - | - | - | - | - | - | - | - |
| Lu et al., 2015 [48] | 76 | 37 | 19:18 | 38.51 ± 13.21 | Healthy Adult Controls | 39 | 22:17 | 38.54 ± 13.15 | Patients with chronic mild traumatic brain injury | - | - | - | - | - | - | - | - |
| Modica et al., 2015 [49] | 112 | 27 | 10:17 | 41.9 ± 10.7 | Healthy demographically matched controls | 85 | 26:59 | 46.0 ± 9.2 | Multiple Sclerosis (MS) Patients | - | - | - | - | - | - | - | - |

(Continued)

Table 3. (Continued)

| Study Reference | Total Participants (n) | Participants (Control group) | | | | Participants (Study group 1) | | | | Participants (Study group 2) | | | | Participants (Study group 3) | | | |
|---|---|---|---|---|---|---|---|---|---|---|---|---|---|---|---|---|---|
| | | n | Gender Ratio (Men:Women) | Mean Age at baseline (years) | Group | n | Gender Ratio (Men:Women) | Mean Age at baseline (years) | Group | n | Gender Ratio (Men:Women) | Mean Age at baseline (years) | Group | n | Gender Ratio (Men:Women) | Mean Age at baseline (Mean years ± SD) | Group |
| Murakami et al., 2018 [50] | 49 | 15 | 10:05 | 74.9±6.2 | Healthy Adult Controls (without iNPH) | 34 | 24:10 | 74.6±5.6 | Patients with iNPH (idiopathic normal pressure hydrocephalus) | - | - | - | - | - | - | - | - |
| Penke et al., 2012 [51] | 143 | 143 | 69:74 | 71.9 ± 0.3 | Healthy Nondemented Adults | - | - | - | - | - | - | - | - | - | - | - | - |
| Pinter et al., 2015 [52] | 69 | - | - | - | - | 17 | 05:12 | 33.1 ± 9.1 | Clinically Isolated Syndrome MS patients | 47 | 18:29 | 35.8 ± 10.5 | Relapsing-Remitting MS patients | 5 | 03:02 | 41.8 ± 9.3 | Secondary Progressive MS patients |
| Qin et al., 2011 [53] | 30 | 15 | 07:08 | 70 | Healthy age and sex matched controls | 15 | 07:08 | 69.8 | Alzheimer's disease patients | - | - | - | - | - | - | - | - |
| Rodrigue et al., 2013 [54] | 113 | - | - | - | - | 113 | 57:76 | 53.96 ± 15.39 | Healthy Adults | - | - | - | - | - | - | - | - |
| Rodrigue et al., 2020 [55] | 166 | - | - | - | - | 166 | 98:68 | 52.75 ± 19.06 | Healthy non-cognitively impaired adults | - | - | - | - | - | - | - | - |
| Salami et al., 2018 [56] | 42 | - | - | - | - | 25 | 12:13 | 36.2 ± 4.4 | Healthy Younger Adults | 17 | 09:08 | 70.1 ± 3.1 | Healthy Older Adults | - | - | - | - |
| Schmalbrock et al., 2016 [57] | 29 | - | - | - | - | 29 | 01:28 | 43.4 ± 9.7 | Relapsing-Remitting Multiple Sclerosis Patients | - | - | - | - | - | - | - | - |
| Smith et al., 2010 [58] | 20 | 5 | - | (Ranging 80–95) | Cognitively and pathologically normal age-matched controls | 4 | - | (Ranging 83–93) | Adults with Pre-clinical AD | 11 | - | (Ranging 74–102) | Adults with mild cognitive impairment | - | - | - | - |
| Steiger et al., 2016 [59] | 62 | - | - | - | - | 31 | 14:17 | 67.3 ± 6.2 | Healthy Elderly Participants | 31 | 17:14 | 24.8 ± 2.8 | Healthy Young Participants | - | - | - | - |
| Sullivan et al., 2009 [60] | 10 | - | - | - | - | 10 | 05:05 | 72.2 | Healthy, non-demented, Elderly Adults | - | - | - | - | - | - | - | - |

(Continued)

Table 3. (Continued)

| Study Reference | Total Participants (n) | Participants (Control group) | | | | Participants (Study group 1) | | | | Participants (Study group 2) | | | | Participants (Study group 3) | | | |
|---|---|---|---|---|---|---|---|---|---|---|---|---|---|---|---|---|---|
| | | n | Gender Ratio (Men: Women) | Mean Age at baseline (years) | Group | n | Gender Ratio (Men: Women) | Mean Age at baseline (years) | Group | n | Gender Ratio (Men: Women) | Mean Age at baseline (years) | Group | n | Gender Ratio (Men: Women) | Mean Age at baseline (Mean years ± SD) | Group |
| **Sun et al., 2017** [61] | 39 | 19 | 15:04 | 65.11 ± 3.71 | Age, gender and education matched controls with Subcortical Ischaemic Vascular Disease (SIVD) but without cognitive impairment | 20 | 16:04 | 63.40 ± 7.98 | SIVD patients with subcortical vascular mild cognitive impairment (svMCI) | - | - | - | - | - | - | - | - |
| **Thomas et al., 2020** [62] | 137 | 37 | 16:21 | 66.1 ± 9.4 | Healthy age matched controls | 100 | 52:48 | 66.4 ± 7.7 | Parkinson's Disease patients within 10 years of diagnosis | - | - | - | - | - | - | - | - |
| **Valdes-Hernandez et al., 2015** [63] | 676 | - | - | - | - | 676 | 372:328 in original sample of 700 | 72.7±0.7 in original sample of 700 | Healthy Elderly Participants | - | - | - | - | - | - | - | - |
| **Van Bergen et al., 2016** [64] | 37 | 22 | 14:08 | 71.91 ± 5.25 | Healthy Adult Controls | 15 | 10:05 | 75.27 ± 7.63 | Adults with MCI | - | - | - | - | - | - | - | - |
| **Wang et al., 2013** [65] | 60 | 18 | 1.43 | 70.52 ± 6.91 | Healthy Age matched controls | 22 | 0.92 | 74.45 ± 8.24 | amnestic mild cognitive impairment patients | 20 | 1.32 | 73.37 ± 9.81 | Alzheimer's Disease patients | - | - | - | - |
| **Yang et al., 2018** [66] | 90 | 30 | 14:16 | 53.17 ± 6.57 | Age, Sex and Education matched healthy adult controls | 30 | 19:11 | 54.97 ± 5.54 | Adults with Type 2 Diabetes Mellitus without Mild Cognitive Impairment | 30 | 12:18 | 55.9 ± 6.54 | Adults with Type 2 Diabetes Mellitus with Mild Cognitive Impairment | - | - | - | - |

**Table 4. Quality scores.**

| Reference | 1 | 2 | 3 | 4 | 5 | 6 | 7 | 8 | 9 | 10 | Quality Score |
|---|---|---|---|---|---|---|---|---|---|---|---|
| Ayton et al., 2019 [27] | 1.00 | 1.00 | 1.00 | 1.00 | 1.00 | 1.00 | 0.50 | 1.00 | 1.00 | 0.50 | 90.00% |
| Bartzokis et al., 2011 [28] | 1.00 | 1.00 | 0.50 | 1.00 | 1.00 | 1.00 | 0.50 | 1.00 | 1.00 | 1.00 | 90.00% |
| Blasco et al., 2014 [29] | 1.00 | 1.00 | 0.50 | 1.00 | 1.00 | 1.00 | 1.00 | 1.00 | 1.00 | 1.00 | 95.00% |
| Blasco et al., 2017 [30] | 1.00 | 1.00 | 0.50 | 0.50 | 1.00 | 1.00 | 1.00 | 1.00 | 1.00 | 0.00 | 80.00% |
| Carpenter et al., 2016 [31] | 1.00 | 1.00 | 0.50 | 1.00 | 1.00 | 1.00 | 0.50 | 1.00 | 1.00 | 1.00 | 90.00% |
| Chen et al., 2018 [32] | 1.00 | 1.00 | 0.00 | 1.00 | 1.00 | 1.00 | 0.50 | 1.00 | 1.00 | 1.00 | 85.00% |
| Darki et al., 2016 [33] | 1.00 | 1.00 | 0.50 | 1.00 | 0.50 | 1.00 | 0.50 | 1.00 | 1.00 | 0.50 | 80.00% |
| Daugherty, 2014 [34] | 1.00 | 1.00 | 0.50 | 1.00 | 1.00 | 1.00 | 1.00 | 1.00 | 1.00 | 1.00 | 95.00% |
| Daugherty, Haacke and Raz, 2015 [35] | 1.00 | 1.00 | 1.00 | 1.00 | 1.00 | 1.00 | 0.50 | 1.00 | 1.00 | 1.00 | 95.00% |
| Daugherty et al., 2019 [36] | 1.00 | 1.00 | 1.00 | 1.00 | 1.00 | 1.00 | 0.50 | 1.00 | 1.00 | 1.00 | 95.00% |
| Ding et al., 2009 [37] | 1.00 | 1.00 | 0.50 | 1.00 | 1.00 | 1.00 | 1.00 | 1.00 | 1.00 | 1.00 | 95.00% |
| Du et al., 2018 [38] | 1.00 | 1.00 | 0.50 | 1.00 | 1.00 | 1.00 | 1.00 | 1.00 | 1.00 | 1.00 | 95.00% |
| Fujiwara et al., 2017 [39] | 1.00 | 1.00 | 0.50 | 1.00 | 1.00 | 1.00 | 0.50 | 1.00 | 1.00 | 1.00 | 90.00% |
| Gao et al., 2017 [40] | 1.00 | 1.00 | 0.50 | 1.00 | 0.50 | 1.00 | 1.00 | 1.00 | 1.00 | 0.50 | 85.00% |
| Ge et al., 2007 [41] | 1.00 | 1.00 | 0.50 | 1.00 | 1.00 | 1.00 | 1.00 | 1.00 | 1.00 | 0.00 | 85.00% |
| Ghadery et al., 2015 [42] | 1.00 | 1.00 | 1.00 | 1.00 | 0.50 | 1.00 | 0.50 | 1.00 | 1.00 | 1.00 | 90.00% |
| Haller et al., 2010 [43] | 1.00 | 1.00 | 1.00 | 1.00 | 0.50 | 1.00 | 1.00 | 1.00 | 1.00 | 1.00 | 95.00% |
| Hect et al., 2018 [44] | 1.00 | 1.00 | 0.50 | 0.50 | 1.00 | 1.00 | 0.50 | 1.00 | 1.00 | 1.00 | 85.00% |
| House et al., 2006 [45] | 1.00 | 1.00 | 0.00 | 1.00 | 1.00 | 1.00 | 1.00 | 1.00 | 1.00 | 0.00 | 80.00% |
| Kalpouzos et al., 2017 [46] | 1.00 | 1.00 | 0.50 | 1.00 | 1.00 | 1.00 | 0.50 | 1.00 | 1.00 | 0.50 | 85.00% |
| Larsen et al., 2020 [1] | 1.00 | 1.00 | 1.00 | 1.00 | 1.00 | 1.00 | 0.50 | 1.00 | 1.00 | 1.00 | 95.00% |
| Li et al., 2015 [47] | 1.00 | 1.00 | 1.00 | 1.00 | 1.00 | 1.00 | 0.50 | 1.00 | 1.00 | 1.00 | 95.00% |
| Lu et al., 2015 [48] | 1.00 | 1.00 | 0.50 | 1.00 | 1.00 | 1.00 | 1.00 | 1.00 | 1.00 | 1.00 | 95.00% |
| Modica et al., 2015 [49] | 1.00 | 1.00 | 1.00 | 1.00 | 1.00 | 1.00 | 1.00 | 1.00 | 1.00 | 1.00 | 100.00% |
| Murakami et al., 2018 [50] | 1.00 | 1.00 | 0.50 | 1.00 | 1.00 | 1.00 | 1.00 | 1.00 | 1.00 | 0.50 | 90.00% |
| Penke et al., 2012 [51] | 1.00 | 1.00 | 1.00 | 1.00 | 1.00 | 0.75 | 1.00 | 1.00 | 1.00 | 1.00 | 97.50% |
| Pinter et al., 2015 [52] | 1.00 | 1.00 | 0.50 | 1.00 | 1.00 | 1.00 | 0.00 | 1.00 | 1.00 | 1.00 | 85.00% |
| Qin et al., 2011 [53] | 1.00 | 1.00 | 0.50 | 1.00 | 1.00 | 1.00 | 1.00 | 1.00 | 1.00 | 0.00 | 85.00% |
| Rodrigue et al., 2013 [54] | 1.00 | 1.00 | 1.00 | 1.00 | 1.00 | 1.00 | 0.50 | 1.00 | 1.00 | 1.00 | 95.00% |
| Rodrigue et al., 2020 [55] | 1.00 | 1.00 | 1.00 | 1.00 | 1.00 | 1.00 | 0.50 | 1.00 | 1.00 | 1.00 | 95.00% |
| Salami et al., 2018 [56] | 1.00 | 1.00 | 0.50 | 1.00 | 0.50 | 1.00 | 0.50 | 0.50 | 1.00 | 1.00 | 80.00% |
| Schmalbrock et al., 2016 [57] | 1.00 | 1.00 | 0.00 | 1.00 | 1.00 | 1.00 | 0.00 | 1.00 | 1.00 | 1.00 | 80.00% |
| Smith et al., 2010 [58] | 1.00 | 1.00 | 0.00 | 0.50 | 0.50 | 1.00 | 1.00 | 1.00 | 1.00 | 0.00 | 70.00% |
| Steiger et al., 2016 [59] | 1.00 | 1.00 | 0.50 | 1.00 | 1.00 | 1.00 | 0.50 | 1.00 | 1.00 | 1.00 | 90.00% |
| Sullivan et al., 2009 [60] | 1.00 | 1.00 | 0.00 | 1.00 | 1.00 | 1.00 | 0.50 | 1.00 | 1.00 | 1.00 | 85.00% |
| Sun et al., 2017 [61] | 1.00 | 1.00 | 0.50 | 1.00 | 1.00 | 1.00 | 1.00 | 1.00 | 1.00 | 1.00 | 95.00% |
| Thomas et al., 2020 [62] | 1.00 | 1.00 | 1.00 | 1.00 | 1.00 | 1.00 | 1.00 | 1.00 | 1.00 | 1.00 | 100.00% |
| Valdes-Hernandez et al., 2015 [63] | 1.00 | 1.00 | 1.00 | 0.50 | 1.00 | 1.00 | 0.50 | 1.00 | 1.00 | 1.00 | 90.00% |
| Van Bergen et al., 2016 [64] | 1.00 | 1.00 | 0.50 | 1.00 | 1.00 | 1.00 | 1.00 | 1.00 | 1.00 | 0.50 | 90.00% |
| Wang et al., 2013 [65] | 1.00 | 1.00 | 0.50 | 1.00 | 1.00 | 1.00 | 1.00 | 1.00 | 1.00 | 0.50 | 90.00% |
| Yang et al., 2018 [66] | 1.00 | 1.00 | 0.50 | 1.00 | 1.00 | 1.00 | 1.00 | 1.00 | 1.00 | 0.00 | 85.00% |

Criteria: (1) Does the study have a clearly defined research objective? (2) Does the study adequately describe the inclusion/exclusion criteria? (3) Is the sample size adequate? (4) Does the study report on the population parameters/demographics? (5) Does the study report detail on appropriate assessment of Cognition? (6) Does the study report detail of the assessment of iron? (7) Does the study provide an appropriate control group? (8) Does the study apply the appropriate statistical analyses? (9) Does the study adequately report the strength of results? (10) Do the authors report on the limitations of their study?

**Table 5. Summary of results.**

| Reference | Summary of Findings relating iron to cognition |
|---|---|
| **Ayton et al., 2019** [27] | Inferior temporal iron levels were increased only in people with clinical diagnosis of dementia who also had moderate (P = 0.0003) and high pathology (P = 0.0190) and fit the CERAD criteria for probable (P = 0.0066) and definite pathology (P = 0.0003), and Braak criteria IV (P = 0.0067) and V (P = 0.0031); In people with high AD pathology, iron was strongly associated (P<0.0001) with the rate of decline in Global Cognition composite; mediation analysis showed that iron levels mediated 17% of the effect of NFTs on Global Cognition; In subjects with low AD pathology, elevated inferior temporal iron burden was associated with decline in global cognitive score (P = 0.001), but not the individual cognitive domain scores |
| **Bartzokis et al., 2011** [28] | Significant negative association between HP iron and episodic memory in men only (p = 0.003); Significant effect of iron genes on association between BG iron and working memory/attention score (p = 0.006); Significant correlation between BG iron and working memory/attention in those without H63D and TfC2 genes (r = -0.49, p = 0.005) |
| **Blasco et al., 2014** [29] | LN R2* values were associated with worse scores in the digit span test (P = 0.011), the ROCF test (P = 0.001), the TMT part A (P = 0.01), and the Iowa Gambling Task test (P = 0.025); Worse performance in the TMT-A were also associated with R2* in CN (P = 0.001) and HS (P = 0.007); HP R2* was associated with worse performance in the ROCF copy test (P = 0.016); HS and HP R2* cut off values discriminate score differences on the deferred memory test (P = 0.039) and the copy ROCF test (P = 0.023), respectively |
| **Blasco et al., 2017** [30] | Increase in R2* negatively correlated with change in visual spatial construction ability and immediate memory (p<0.05); Copy memory scores were inversely associated with R2* at the L-CN (r = 20.409; P = 0.034), L- and R- PA (r = 20.383; P = 0.048 and r = 20.524; P = 0.005, respectively), and R-PU (r = 20.575; P = 0.002); Immediate and deferred memory scores were inversely associated with R2* at the R-TH (r = 20.403; P = 0.037 and r = 20.395; P = 0.041); Worse TMT-A scores were associated with increased R2* at R- and L-PA (r = 0.440; P = 0.024 and r = 0.529; P = 0.005) |
| **Carpenter et al., 2016** [31] | Significant positive association between mean iron in basal ganglia and spatial IQ (p = 0.02); Iron in the R-CN (p<0.01), L-CN (p<0.05) and SN (p<0.05) had significant positive association with spatial IQ, but only R-CN relationship was withheld after correction for multiple comparisons; No association between spatial IQ and iron in GP, PU or TH |
| **Chen et al., 2018** [32] | Significant correlation between brain iron in GP and fluid composite score in control group (p<0.01); Baseline PU brain iron is negatively associated with changes in oral reading recognition test scores in the control group (p<0.01) |
| **Darki et al., 2016** [33] | Significant correlation between CN iron working memory performance in children (r = 0.64, p = 0.004) and adults (r = 0.46, p = 0.04); mainly driven by the R-CN in children; Other subcortical nuclei were not significantly correlated to working memory performance after Bonferroni correction for multiple comparisons |
| **Daugherty, 2014** [34] | Greater iron content at baseline was associated with slower iron accumulation in CN (p<0.05) and PU (p = 0.05); Higher metabolic syndrome score was associated with higher iron in the CN (p = 0.003) and LQ (p = 0.02); Inflammation score was unrelated to iron content; Non-verbal working memory didn't change with age (p = 0.76); Verbal working memory improved over two years (p<0.001); Virtual Morris water maze test score was unrelated to iron or volume in any region |
| **Daugherty, Haacke and Raz, 2015** [35] | Cognitive switching ability was found to be inversely proportional to striatal iron (p<0.001) |
| **Daugherty et al., 2019** [36] | Greater baseline CN iron was associated with lesser improvement in working memory over 2 years (p = 0.01); Change in verbal working memory was unrelated to iron in the PU (p>0.52) or HP (p>0.17); Episodic memory wasn't associated with baseline iron (p>0.31) |
| **Ding et al., 2009** [37] | Mean MMSE score was significantly lower in AD patients than controls (p<0.001); AD group showed significantly lower phase value in all brain structures measured (p<0.05); Phase value in R-head of HP had positive correlation with MMSE score (r = 0.603, p = 0.000) and negative correlation with disease duration (r = -0.677, p = 0.013) |

*(Continued)*

**Table 5.** (Continued)

| Reference | Summary of Findings relating iron to cognition |
|---|---|
| Du et al., 2018 [38] | Bilateral CN and PU susceptibility values are significantly higher in AD patients than controls (p<0.05); Bilateral RN susceptibility was significantly lower in AD patients than controls (p<0.05); left CN susceptibility is correlated with a decrease in MMSE score (p<0.01) and MoCA score (p<0.05) |
| Fujiwara et al., 2017 [39] | Cognitive z scores were negatively associated with GP QSM (p = 0.03); No other QSM scores were correlated with cognition; Cognitive z score was (non significantly) related to GP R2* (p = 0.099); Controls showed no significant relationships between iron measures and cognition |
| Gao et al., 2017 [40] | MMSE and MoCA was significantly higher in AD than both MCI and controls and was significantly higher in MCI than controls (p<0.05); L-DN, L-CN, PU of MCI group had significantly lower phase than controls; DN, R-RN, PU of AD group had significantly lower phase than MCI group; HP, DN, RN, CN, GP, PU and L-TC phase in AD group were significantly lower than controls; Lower Phase was significantly correlated with higher brain iron concentration (p<0.05) |
| Ge et al., 2007 [41] | Iron was significantly higher in MS patients than controls in GP (p = 0.007), PU (p = 0.002), TH (p = 0.03); Significant correlation between Magnetic Field Correlation for iron (MFC) value in the TH and the CVLT test performance (r = -0.42, p = 0.04) and RCFT performance (r = -0.50, p = 0.03); MFC in the PU correlated DSB test performance (r = 0.45, p = 0.03) |
| Ghadery et al., 2015 [42] | Higher age associated with lower education level, higher frequency of risk factors, worse cognitive performance, greater extent of focal brain lesions and lower brain volume (p<0.05); Higher iron load in PA related inversely with all cognitive measures except memory; R2* in PU was related to global cognitive function and psychomotor speed (p<0.05); No relationship between R2* in neocortex or HP and cognition; Associations between R2* iron and cognition were strongest in ages above 71; R2* iron in the pallidum accounted for 9% of the age-related variance in executive function, 7% in global cognitive function, and 8% in psychomotor speed; R2* iron in the PU accounted for 24% of the age-related variance in executive function, 18% in global cognitive function, and 21% in psychomotor speed |
| Haller et al., 2010 [43] | There was a significantly increased iron concentration in R-PA and R-SN in MCI groups compared to controls (p<0.01); There was significantly decreased iron concentration in the R-RN in MCI groups compared to controls (p<0.05); No difference in iron concentration was found in any regions between stable and progressive MCI |
| Hect et al., 2018 [44] | Brain iron in CN (p = 0.03), PU (p<0.01), GP (p = 0.04) and SN (p<0.01) correlated with general intelligence scores; Brain iron in the CN (p<0.001) and PU (p<0.01) correlated processing speed; HP (p>0.69) and RN (p>0.33) iron content were unrelated to cognition; Greater general brain iron content predicted faster processing speed (p = 0.02) and better general intelligence (p = 0.01) |
| House et al., 2006 [45] | Least cognitively impaired memory-complaint group (MC1) had significantly higher R2 in R-temporal cortex and significantly lower R2 in the L-internal capsule compared to controls; MC1 and MC2 groups showed significant correlation between R2 and immediate, short-delay and long-delay free recall scores in CVLT in TH and RN (r = -0.62 to -0.77, p<0.04); R2 in the RN was negatively correlated to MMSE scores (p<0.02); Negative correlation coefficients were more frequently associated with R2 in GM regions for the immediate free recall scores (p = 0.001), SDFR cognitive score (p = 0.0002) and LDFR test scores (p = 0.0002) |
| Kalpouzos et al., 2017 [46] | Higher striatal iron in the older group was associated with poorer recall in motor condition (p = 0.02); Striatal iron was not significantly associated with recall in the younger adults (p>0.7); Bootstrapping analysis indicated reliable association between striatal R2* and memory performance in older group; Greater striatal iron was associated with less inferior frontal cortex activation when age and striatal volume were controlled for (p = 0.05); Higher iron in R-PU was associated with lower activity in the R-PU when controlling for age and R-PU volume (p = 0.04) |
| Larsen et al., 2020 [1] | Developmental trajectory of R2* in PU significantly interacted with overall cognitive score (p = 0.006) with poorer performance becoming increasingly associated with lower R2* levels; Developmental trajectories of R2* were most strongly associated with complex cognitive performance (p = 0.004) with significant association between R2* trajectory and social cognition (p = 0.031) and executive function (p = 0.032), No significant effect of R2* on memory performance (p = 0.39) |

*(Continued)*

**Table 5.** (Continued)

| Reference | Summary of Findings relating iron to cognition |
|---|---|
| **Li et al., 2015** [47] | Decrease in manual dexterity score was significantly associated with increase in magnetic susceptibility in the GP and RN; In younger participants the susceptibility-dexterity correlation was significant for GP ($p<0.01$) but not RN ($p = 0.028$); In older participants the susceptibility-dexterity correlation was significant for RN ($p<0.05$) but not for GP ($p = 0.11$); Only GP magnetic susceptibility was a significant predictor of variance in manual dexterity score (with higher GP magnetic susceptibility associated with lower manual dexterity score) |
| **Lu et al., 2015** [48] | Compared to control group, cmTBI patients had significantly higher angle radian values in CN ($p<0.001$), LN ($p<0.001$), L-HP ($p<0.05$), R-HP ($p<0.001$), L-RN ($p<0.05$), R-RN ($p<0.001$), R-SN ($p<0.001$), splenium of CC ($p<0.005$); Cognitive score in the patient group were negatively correlated to angle radian values in the R-SN ($r = -0.685$, $p<0.001$) |
| **Modica et al., 2015** [49] | MS patients significantly more cognitively impaired then healthy controls; Mean phase significantly lower in patients with MS in TH, CN, PL; Mean phase of CN, PU, GP and PL but not TH correlated cognitive test scores when volume was controlled for ($p<0.05$) |
| **Murakami et al., 2018** [50] | 3 months after shunt surgery, brain-type Tf strongly correlated with MMSE scores ($r = 0.697$, $p = 0.037$) and FAB score ($r = 0.727$, $p = 0.041$); 12 months after shunt surgery, brain-type Tf moderately correlated MMSE scores ($r = 0.549$, $p = 0.022$) and FAB score ($r = 0.373$, $p = 0.154$); mRS scores were not associated with brain-type Tf before or after surgery |
| **Penke et al., 2012** [51] | Compared with the group without detectable Iron Deposits (IDs), those with IDs at age 72 had significantly lower general cognitive ability at age 70 ($p = 0.043$), and age 72 ($p = 0.0004$), but not at age 11 ($p = 0.19$); Censored correlations showed greater IQ at 11 was significantly associated with fewer iron deposits at age 72 ($p = 0.0324$, $r = -0.19$); Reading recognition tests showed significant negative association with iron deposits ($r = -0.18$, $p = 0.0253$); Iron deposits were significantly associated with lower general cognitive ability at age 70 ($r = -0.27$, $p = 0.0015$) and 72 ($r = -0.31$, $p<0.0001$) |
| **Pinter et al., 2015** [52] | Magnetisation transfer ration for normal appearing brain tissue explained 26.7% variance in overall cognition; Overall iron deposition did not account for variance in overall cognition significantly; Basal ganglia R2* explained 22.4% variance of cognitive efficiency; HP magnetic transfer ration of normal appearing brain tissue (22.4%) also accounted for memory variance; TH volume was the only predictor of memory function after multivariate modelling; The only predictor of cognitive efficiency after multivariate modelling was R2* in the basal ganglia (explaining 22.4% variance) |
| **Qin et al., 2011** [53] | R2* in HP, PC, PU and CN of AD significantly higher than control group ($p<0.05$); R2* in PC, HP and L-PU in mild AD group were significantly higher than in controls ($p<0.05$); R2* in HP, PC, PU and DN in patients with severe AD were significantly higher than the control and mild AD groups; MMMSE was negatively correlated with R2* and iron concentration in PC and HP in AD group ($p<0.01$) |
| **Rodrigue et al., 2013** [54] | Increased HP iron and smaller HP volume accounted for age-related memory deficits ($p = 0.05$) whereas, CN did not have this effect; Younger participants with larger HP and lower HP iron had the highest memory composite scores; Single indirect path modelling showed a negative indirect association of age with HC volume through increased HP iron concentration ($p<0.0001$) and advanced age was indirectly related to poorer memory performance through a shorter HP T2* and then smaller HP volume ($p<0.0001$) |
| **Rodrigue et al., 2020** [55] | Significant decline in performance across all levels of n-back tests ($p<0.05$) with increasing age but no iron interaction in this model ($p>0.174$); No association found between iron and performance in executive function performance |

(*Continued*)

Table 5. (Continued)

| Reference | Summary of Findings relating iron to cognition |
|---|---|
| **Salami et al., 2018** [56] | Significant negative association between striatal R2* and coherence in connectivity of the CN (r = -0.41, p = 0.008) and PU (r = -0.32, p = 0.047); Significant association between striatal iron and coherence of connectivity in the CN resting-state network in the older group (r = -0.53, p = 0.04) but not in the younger group (r = -0.24, p = 0.27); Association between QSM and CN connectivity coherence confirmed significance (r = 0.398, p = 0.015) but the PU connectivity coherence was not significantly associated with QSM (p = 0.07); Significant positive association between coherence of PU networks and task performance with the dominant hand across age groups (p = 0.04); Significant association between striatal iron and motor performance with the dominant hand across the age groups (p = 0.047) |
| **Schmalbrock et al., 2016** [57] | Flanker test for inhibitory control was significantly associated with QSM in CN (p = 0.01) and anterior PU (p = 0.045); Stroop test for inhibitory control was not significantly associated with brain iron measures; Disease duration was significantly associated with QSM in the CN (p = 0.02); Sqrt (Flanker) was significantly associated with age adjusted QSM in the CN (p = 0.0058) and anterior PU (p = 0.016); Duration adjusted Expanded disability status score was significantly associated with age adjusted QSM in the posterior PU (p = 0.032) and age adjusted R2 in the CN (p = 0.014), PU (p = 0.0059, Anterior p = 0.0054, Posterior p = 0.019) |
| **Smith et al., 2010** [58] | Controls had significantly lower cortical redox iron than other groups (p<0.05); Controls had significantly less iron accumulation in the cerebellum but had high metal deposition in the purkinje cell layer; Iron accumulation did not occur not in purkinje cells for MCI brains but instead in spherical glial associated structures; MCI cases had significantly more iron accumulation than controls in the purkinje layer associated with glial cells |
| **Steiger et al., 2016** [59] | In ventral striatum there was a positive correlation between VLMT learning performance and Magnetic transfer (MT), but a negative correlation between VLMT recognition performance and R2*; VLMT learning performance was predicted by the ratio of MT/R2* |
| **Sullivan et al., 2009** [60] | Higher iron in CN predicted lower dementia rating scale score (r = -0.7, p = 0.0232; Rho = -0.56, p = 0.0944); Lower arithmetic score correlated higher iron in CN (r = −0.64, p = 0.0481; Rho = −0.70, p = 0.0359) and putamen (r = −0.78, p = 0.0077; Rho = −0.65, p = 0.0495); TH iron was predictive of Digit Symbol output (r = 0.77, p = 0.0088; Rho = 0.57, p = 0.0865), time taken to complete the test (r = −0.79, p = 0.0069; Rho = −0.56, p = 0.0909), and MMSE scores (r = 0.66, p = 0.0397; Rho = 0.47, p = 0.1611); In the two choice test CN iron correlated with longer reaction time by the left (r = 0.56, p = 0.0918) and right (r = 0.79, p = 0.0062) hands, higher GP iron correlated with longer reaction time by the right hand (r = 0.65, p = 0.0421) and higher PU iron correlated with longer movement time by the left (r = 0.70, p = 0.024) hand; Fine finger movement speed showed no significant relationship with iron estimates in any region; In the Digit Symbol grid, CN and TH iron accounted for 80% of the variance; Low TH iron (p = 0.0096) was a unique predictor of performance over the caudate iron measure (p = 0.5192) |
| **Sun et al., 2017** [61] | svMCI group had significantly lower composite, attention-executive, memory and language z scores than controls; significantly higher susceptibility in svMCI group over controls in R-HP (p<0.01), L-HP (p<0.01), R-PU (p<0.05); svMCI group had significantly negative correlation between sus in R-HP and memory z sore (p = 0.012); susceptibility in R-HP of svMCI group was positively correlated to language z score (p = 0.026); susceptibility in R-PU in the svMCI group was significantly negatively correlated to attention-executive z score (p = 0.033); composite z score not related to susceptibility |
| **Thomas et al., 2020** [62] | Increase in QSM in PD compared to controls in prefrontal cortex, R-PU and R-temporal cortex (p<0.05); Increased QSM in SN in PD compared to controls (p = 0.004); In PD patients there was susceptibility increase with decreasing MoCA scores in HP, TH, CN, caudal regions of ventromedial prefrontal cortex, regions of basal forebrain, R-PU and R-insular cortex; Increased absolute susceptibility with increased dementia risk score in PD patients (p<0.05); widespread QSM increases in patients with poor visual performance (p<0.05); PD group showed significant increase in susceptibility (p<0.05) with UPDRS- III in right PU |

(*Continued*)

**Table 5.** (Continued)

| Reference | Summary of Findings relating iron to cognition |
|---|---|
| **Valdes-Hernandez et al., 2015** [63] | All 3 cognitive factors (Memory, Processing Speed and Fluid intelligence) were significantly negatively correlated with total Iron deposition (r = -0.165) at older age, even when controlling for all other health factors; No significant correlation between Iron deposition and cognition at 11 y/o |
| **Van Bergen et al., 2016** [64] | MCI and healthy controls differed significantly in MoCA, VMLT, BNT and WMS cognitive tests; Strong significant increase in susceptibility in APOe4 carriers of the MCI group in CN (p<0.001) and frontal, temporal, parietal and occipital cortices (p<0.001) |
| **Wang et al., 2013** [65] | Regions where MMSE score was significantly correlated to angle radian values were the R&L-cerebellar hemisphere, R&L-HP, R&L-RN, R-CN, R&L-LN, R&L-TH, and splenium of CC, where correlation coefficients were 0.36999, 0.3783, 0.40081, 0.40741, 0.2892, 0.2599, 0.2593, 0.40462, 0.26039, 0.54453, 0.46979, -0.28888 (P values = 0.00362, 0.00288, 0.00151, 0.00123, 0.02501, 0.04492, 0.04543, 0.00134, 0.0445, <0.001, <0.001, 0.02519, respectively) |
| **Yang et al., 2018** [66] | T2DM without MCI group had increased susceptibility in bilateral CN, HP, left PU and right SN compared to controls (p<0.05); T2DM with MCI group had significantly increased susceptibility in right CN, SN and left PU compared to T2DM without MCI group (p<0.05); Susceptibility values for right CN, SN and left PU were closely correlated to cognitive scores (r>-0.55, p<0.04) |

CN = Caudate Nucleus, GP = Globus Pallidus, PU = Putamen, SN = Substantia Nigra, HP = Hippocampus,
RN = Red Nucleus, DN = Dentate Nucleus, TH = thalamus, PA = Pallidum, AM = amygdala, WM = White Matter,
PL = pulvinar nucleus of the thalamus, CC = corpus collosum.

(including memory, general intelligence, visual performance, processing speed, social cognition and BOLD modulation). Every other study reviewed reported significant associations between iron levels in specific brain regions and individual measures of cognition, as presented in Fig 2.

## Discussion

### Summary of evidence

This review analysed human studies in which brain iron and cognition were measured and their relationship assessed statistically. Many of studies assessed reported a significant relationship between total brain iron and general cognitive performance and many links between regional iron levels and specific measures of cognition were also reported. Memory function was the most frequently reported cognitive measure to be correlated with brain iron, however, this was the most frequently assessed cognitive outcome. Regions where iron was most frequently reported to correlate with memory performance were the Caudate nuclei, Hippocampus and Thalamus. All other regions were also associated with memory in at least one study except for the Globus Pallidus where regional iron had no reported associations with memory. The associations between the caudate, hippocampus and thalamus iron and memory are somewhat unsurprising as each of these regions are known to be involved in different facets of memory function [67–69] and so it is plausible that disruption of these circuits via iron accumulation would confer memory dysfunction. The efficacy of interactions between the caudate and hippocampus in memory function has been associated with availability of dopamine receptors [70, 71], which has in turn been proposed as having a potential role in iron accumulation [72]. Studies have suggested that iron and dopamine can interact to induce oxidative stress and neurodegeneration by forming a toxic couple [72]. Animal studies have also

| Region of Interest | General Cognition/IQ | Processing Speed | Memory | Visuospatial abilities | Executive function | Connectivity coherence | Inhibitory control | Disability Scores | Motor Function | Reaction times |
|---|---|---|---|---|---|---|---|---|---|---|
| Caudate Nuclei | 4 | 1 | 8 | 0 | 0 | 1 | 1 | 1 | 0 | 1 |
| Putamen | 7 | 1 | 2 | 0 | 1 | 1 | 1 | 2 | 1 | 1 |
| Hippocampus | 0 | 0 | 8 | 0 | 0 | 0 | 0 | 0 | 0 | 0 |
| Thalamus | 2 | 0 | 6 | 0 | 0 | 0 | 0 | 0 | 0 | 1 |
| Globus Pallidus | 4 | 0 | 0 | 0 | 0 | 0 | 0 | 0 | 1 | 1 |
| Striatum | 2 | 0 | 2 | 0 | 0 | 1 | 0 | 0 | 2 | 0 |
| Basal Ganglia | 2 | 1 | 1 | 0 | 0 | 0 | 0 | 0 | 0 | 0 |
| Red Nuclei | 0 | 0 | 2 | 0 | 0 | 0 | 0 | 0 | 1 | 0 |
| Substantia Nigra | 3 | 0 | 1 | 0 | 0 | 0 | 0 | 0 | 0 | 0 |
| Pallidum* | 1 | 0 | 1 | 0 | 0 | 0 | 0 | 0 | 0 | 0 |
| Lenticular Nucleus | 1 | 0 | 2 | 0 | 0 | 0 | 0 | 0 | 0 | 0 |
| Temporal Coretx | 1 | 0 | 1 | 0 | 0 | 0 | 0 | 0 | 0 | 0 |
| Hypothalamus | 0 | 0 | 1 | 0 | 0 | 0 | 0 | 0 | 0 | 0 |
| Cerebellum | 0 | 0 | 1 | 0 | 0 | 0 | 0 | 0 | 0 | 0 |

**Fig 2. Regional associations between iron and cognition.** Figure presents number of studies reporting significant association (p<0.05) between regional iron and cognition measures. *Pallidum had associations between regional iron and memory in one study [30] but had association in all cognitive measures except memory in a second study [42].

demonstrated that iron deficient mice and rats show decreased dopamine transporter and receptor levels and general dopaminergic dysfunction [73, 74]. This suggests that with an increase iron, there could be an increase in dopamine receptors and transporters, enhancing toxic coupling between iron and dopamine and thus increasing neurodegeneration in dopamine rich regions, however, this requires further investigation.

Furthermore, higher iron levels in the caudate nuclei were also consistently reported to correlate to poorer general cognitive performance. However, the putamen had the most reported associations with general cognition, with the Globus Pallidus and the Substantia Nigra also being associated with general cognition in more than one study. The putamen has roles in many different neurological functions such as, sensory and motor information processing, learning and language [75–77]. This could explain the consistency of reports that iron accumulation here is associated with poorer general cognitive performance, further suggesting that iron accumulation causes atrophy which leads to a localised disruption of function.

Although assessed in less of the studies reviewed, there were associations between reduced motor function and increased striatal iron content, as well as, increased iron in the Putamen and increased disability scores, such as, Dementia rating scale, Extended disability status score and the UPDRS-III for rating of Parkinson's pathology. Due to its many neurological roles and connections, atrophy in the Putamen is known to be involved in pathology of several diseases such as, Parkinson's disease, Multiple Sclerosis and Dementia with Lewy Bodies [78–81]. The striatum consists of the caudate and putamen and is crucial for connections to the basal ganglia which is heavily involved in motor control [82]. These associations therefore suggest that iron accumulation is capable of either causing atrophy or is accompanied by atrophy, which in turn causes regional damage that can interfere with circuitry in the brain. This is in line with the findings of several of the included studies that increases in regional brain iron were strongly associated with regional volume decrease [34, 35, 49, 54], suggesting that brain iron increase is correlated with atrophy.

Whilst not evaluated in the included studies, differences in iron status have been observed between sexes particularly during development. Larsen et al. observed these differences in their 2020 study, which determined that male brain iron levels plateau at an earlier age than in females in some brain regions. Due to this later plateau during development females generally begin adulthood with higher brain iron levels than men. However, at older age, females are shown to have generally lower iron stores in some brain regions than males, potentially due to menstruation [83–85]. Female brain iron deficit mediated by menstruation would however, be highly variable, dependent on the characteristics of an individuals' menstruation (i.e. menstruation pattern, heaviness of blood loss etc.) [83]. This may put females at a lower risk of brain iron-mediated cognitive impairment, however the effects of sex-mediated brain iron on cognition have not been extensively studied.

All studies included in this review controlled for sex during their analyses. Fifteen of the included studies assessed sex-mediated brain iron differences statistically; 1 study found that while temporal iron levels did not significantly differ between men and women, cerebellar iron was significantly higher in males compared to females [27]; 1 studies found significantly higher hippocampal iron in men compared to women [28]; 1 study found that regional brain iron in women plateaus later than in men during development [1] and the remaining 12 studies observed no significant difference in brain iron in any assessed region between males and females [29, 31, 34–36, 38, 42, 44, 47, 51, 54, 63].

Although primarily focusing on the associations between brain iron and cognition, some of the papers reviewed did provide insight into potential mechanisms for this relationship. It has been previously reported that iron in the brain tends to localise to protein aggregates and some studies have shown that iron plays a role in the toxicity of some of these aggregates [86–89]. In fact, when amyloid β (Aβ) is complexed with iron it can induce ROS via Fenton's reaction leading to oxidative stress and activation of the Bcl-2 apoptotic pathway [18, 19]. Iron has also been shown to localise with protein aggregates such as tau and amyloid beta in animal models for AD and PD [88, 90, 91]. Several of the studies included in this review reported that iron was localised to Aβ plaques and neurofibrillary tangles [27, 58]. A study by Ayton et al. [27], included in this review, found that brain iron level mediated 17% of the effect of Neurofibrillary tangles on cognitive performance. This, taken with the afore mentioned literature, suggests that iron could *amplify* neurodegenerative processes such as protein misfolding, rather than being a primary cause or effect of disease.

## Limitations

Whilst this article was able to provide a comprehensive review of the literature investigating the relationship between brain iron and cognition, there were several limitations to this study.

Firstly, there was a wide variety of methods for measuring both brain iron and cognition and this must be considered when comparing the included studies. Secondly, although a thorough search of the literature was conducted, it is possible that relevant studies were missed and thus not included. Furthermore, all included studies were published articles or theses and thus there is an element of publication bias in this review that must be considered. Additionally, some of the studies included in this review had relatively small sample sizes which may reduce the power of some of the conclusions made. The participants all bar one of the studies in this review were adults and so the findings cannot be applied to children or adolescents. Finally, the potential mechanisms by which iron accumulation in the brain could cause cognitive dysfunction were not assessed in this review and remain unclear.

## Conclusions

To conclude, this review has investigated the effects of brain iron on aspects of cognition. There is consistent evidence in the studies reviewed that in adulthood, an increase in brain iron had a detrimental effect on cognitive ability. However, it seems that iron accumulates heterogeneously across brain regions and when only some regions have high iron, their specific function can be impaired. In this way, increased iron in the Caudate nuclei, Hippocampus and Thalamus is consistently reported to correlate to poorer memory performance; whereas, increased iron in the putamen was more consistently reported to correlate to poorer general cognition. These findings strongly suggest an effect of brain iron on cognition and this relationship should therefore be investigated further. Going forward, it is important to determine whether iron is a primary cause of brain atrophy or whether brain iron accumulation is a secondary effect of brain atrophy. Regardless of the mechanisms underlying the relationship between brain iron and cognition, MRI techniques for quantifying brain iron therefore show promise as a potential non-invasive biomarker for age-related cognitive decline.

## Supporting information

**S1 Checklist. PRISMA 2009 checklist.**
(DOCX)

## Acknowledgments

The authors would like to thank the Roland Sutton Academic Trust for their financial support.

## Author Contributions

**Conceptualization:** Chris J. McNeil, Gordon D. Waiter.

**Data curation:** Holly Spence.

**Formal analysis:** Holly Spence.

**Funding acquisition:** Gordon D. Waiter.

**Investigation:** Holly Spence.

**Project administration:** Holly Spence, Chris J. McNeil, Gordon D. Waiter.

**Supervision:** Chris J. McNeil, Gordon D. Waiter.

**Validation:** Chris J. McNeil, Gordon D. Waiter.

**Visualization:** Holly Spence.

**Writing – original draft:** Holly Spence.

**Writing – review & editing:** Holly Spence, Chris J. McNeil, Gordon D. Waiter.

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
