## [Decision Letter · Decision Letter 0]

24 Aug 2020

PONE-D-20-20440

The impact of brain iron accumulation on cognition: A systematic review

PLOS ONE

Dear Dr. Spence,

Thank you for submitting your manuscript to PLOS ONE. After careful consideration, we feel that it has merit but does not fully meet PLOS ONE’s publication criteria as it currently stands. Therefore, we invite you to submit a revised version of the manuscript that addresses the points raised during the review process.

Although your manuscript was well-received some minor clarifications and corrections are in order. Please thoroughly address all points below.

We look forward to receiving your revised manuscript.

Kind regards,

Efthimios M. C. Skoulakis, PhD

Academic Editor

PLOS ONE

Journal Requirements:

Reviewers' comments:

Reviewer's Responses to Questions

**Comments to the Author**

1. Is the manuscript technically sound, and do the data support the conclusions?

Reviewer #1: Yes

2. Has the statistical analysis been performed appropriately and rigorously? 

Reviewer #1: N/A

3. Have the authors made all data underlying the findings in their manuscript fully available?

Reviewer #1: Yes

4. Is the manuscript presented in an intelligible fashion and written in standard English?

Reviewer #1: Yes

5. Review Comments to the Author

Reviewer #1: #Referee

Comments to the Corresponding Author

PONE-D-20-20440

The authors have invested in a systematic approach to compare and evaluate the current literature and explore the relationship between brain iron and cognition. Identifying that iron appears to accumulate heterogeneously in regions of the brain and when only a few regions have a high iron content, its specific function may be deficient. Thus, increase iron in the caudate nuclei, hippocampus and thalamus is correlated with lower memory performance; whereas, increase iron in the putamen has been reported more consistently to correlate with poorer general cognition. The data presented here is very interesting, and the study design was generally successful.

However, the following weaknesses and improvements must be considered:

Point 1. However, according to Bartzokis et al. (2007) ferritin iron levels are higher in men than in women, which may contribute to the risk of developing neurodegenerative diseases at an early age. In this review, the gender factor was not discussed, according to the studies consulted, there would be changes in iron levels in women? And what would be the average age of both sexes attributed to cognitive impairment due to high levels of iron?

Point 2. It is suggested that there is more work /reflection on how high levels of iron act on the nervous system and cause cognitive alterations.

Line-By-Line

Point 3. Line 41- “In the healthy adult brain”. Please specify whether the quote refers to humans?

Point 4. Line 54- define ROS before using the acronym.

Point 5. Line 56- What damage excessive ROS generation can cause at the brain level, how neurodegeneration can occur via Fenton Reaction?

Point 6. Line 84- PD e AD previously abbreviated.

Point 7. Line 89- The systematic review brings gaps to be answered and hypotheses to be raised, the addition of a hypothesis to the study is encouraged.

Point 8. Line 115- define HS before using the acronym.

Point 9. Line 142- In this sentence it indicates that 43 studies were used, however, in Table 2 and Table 3 there is a citation of 41 studies.

Point 10. Line 143- Figure 1 is somewhat unreadable.

Point 11. Line 149 and 157- Standardize the use of abbreviations in tables.

Point 12. Line 126-127- Previously abbreviated.

Point 13. Line 237- define Aβ before using the acronym.

Point 14. Line 253-“all included studies were published articles or theses”. It is necessary to insert in the material and methods the use of theses in this review.

Point 15. Line 265-267- The insertion of a graphic scheme is strongly encouraged, making these possible correlations.

6. PLOS authors have the option to publish the peer review history of their article (what does this mean?). If published, this will include your full peer review and any attached files.

Reviewer #1: No

---

## [Author Response · Author response to Decision Letter 0]

9 Sep 2020

Dear Sir/Madam,

Manuscript title: The impact of brain iron accumulation on cognition: A systematic review

We thank the reviewers for their time and comments on the above titled manuscript. We have edited the manuscript to address these comments and include a revised submission with these revisions highlighted for your use. Below we have detailed how we have addressed each point made in the referee’s decision letter and hope that the revised manuscript now meets PLOS ONE’s publication criteria.

“Point 1. However, according to Bartzokis et al. (2007) ferritin iron levels are higher in men than in women, which may contribute to the risk of developing neurodegenerative diseases at an early age. In this review, the gender factor was not discussed, according to the studies consulted, there would be changes in iron levels in women? And what would be the average age of both sexes attributed to cognitive impairment due to high levels of iron?” An additional paragraph has been added into the discussion to address the current evidence surrounding sex-mediated brain iron differences. Statistical assessments of gender effect on brain iron in the studies included for review are also detailed in this paragraph. See lines 239-255

“Point 2. It is suggested that there is more work /reflection on how high levels of iron act on the nervous system and cause cognitive alterations.” More has been specified regarding the mechanisms by which iron can induce and amplify oxidative damage and eventually lead to neurodegeneration. See introduction section.

“Point 3. Line 41- “In the healthy adult brain”. Please specify whether the quote refers to humans?” This specification has now been made in line 41.

“Point 4. Line 54- define ROS before using the acronym.” This abbreviation has now been defined upon first mention in line 54.

“Point 5. Line 56- What damage excessive ROS generation can cause at the brain level, how neurodegeneration can occur via Fenton Reaction?” We have added extra information to this paragraph to further outline the role of ROS and Fenton’s reaction in neurodegeneration. See lines 54-61.

“Point 6. Line 84- PD e AD previously abbreviated.” These have now been abbreviated. See line 88.

“Point 7. Line 89- The systematic review brings gaps to be answered and hypotheses to be raised, the addition of a hypothesis to the study is encouraged.” A key hypothesis has now been defined in lines 92-93 of the concluding introductory paragraph.

“Point 8. Line 115- define HS before using the acronym.” This has now been defined as the authors full name, rather than using initials. See line 120.

“Point 9. Line 142- In this sentence it indicates that 43 studies were used, however, in Table 2 and Table 3 there is a citation of 41 studies.” and “Point 10. Line 143- Figure 1 is somewhat unreadable.”

This point has highlighted an error in this sentence and in the flow chart of figure 1. The sentence mentioned has now been changed and a new version of figure 1, also altered to improve readability has been submitted with this resubmission.

 “Point 11. Line 149 and 157- Standardize the use of abbreviations in tables.” Abbreviations in all tables have now been standardised.

“Point 12. Line 126-127- Previously abbreviated.” We were unable to determine what this point was referring to, but have checked abbreviations between line 157 and 237.

“Point 13. Line 237- define Aβ before using the acronym.” This has now been rectified in line 262.

“Point 14. Line 253-“all included studies were published articles or theses”. It is necessary to insert in the material and methods the use of theses in this review.” This has now been specified in lines 121-122 of the Methods section of this submission.

“Point 15. Line 265-267- The insertion of a graphic scheme is strongly encouraged, making these possible correlations.” Figure 2 shows these correlations in a graphic scheme and has been resubmitted with the paper in a higher quality format. 

Yours Faithfully,

Holly Spence

Post Graduate Researcher

Direct Phone: 07519122260

Email: h.spence.19@abdn.ac.uk

---

## [Editor Report · Decision Letter 1]

1 Oct 2020

The impact of brain iron accumulation on cognition: A systematic review

PONE-D-20-20440R1

Dear Holly Spence,

We’re pleased to inform you that your manuscript has been judged scientifically suitable for publication and will be formally accepted for publication once it meets all outstanding technical requirements.

Kind regards,

Efthimios M. C. Skoulakis, PhD

Academic Editor

PLOS ONE
---

## [Editor Report · Acceptance letter]

2 Oct 2020

PONE-D-20-20440R1 

The impact of brain iron accumulation on cognition: A systematic review 

Dear Dr. Spence:

I'm pleased to inform you that your manuscript has been deemed suitable for publication in PLOS ONE. Congratulations! Your manuscript is now with our production department. 

Kind regards, 

on behalf of

Dr. Efthimios M. C. Skoulakis 

Academic Editor

PLOS ONE